# Development and Evaluation of Solid Lipid Nanoparticles for the Clearance of Aβ in Alzheimer’s Disease

**DOI:** 10.3390/pharmaceutics15010221

**Published:** 2023-01-09

**Authors:** Meghana Goravinahalli Shivananjegowda, Umme Hani, Riyaz Ali M. Osmani, Ali H. Alamri, Mohammed Ghazwani, Yahya Alhamhoom, Mohamed Rahamathulla, Sathishbabu Paranthaman, Devegowda Vishakante Gowda, Ayesha Siddiqua

**Affiliations:** 1Department of Pharmaceutics, JSS College of Pharmacy, JSS Academy of Higher Education and Research, Mysuru 570015, India; 2Department of Pharmaceutics, College of Pharmacy, King Khalid University, Abha 61421, Saudi Arabia; 3Department of Cell Biology and Molecular Genetics, Sri Devaraj Urs Medical College, Sri Devaraj Urs Academy of Higher Education and Research, Kolar 563101, India; 4Department of Pharmaceutics, Cauvery College of Pharmacy, Mysuru 570028, India; 5Department of Clinical Pharmacy, College of Pharmacy, King Khalid University, Abha 62529, Saudi Arabia

**Keywords:** drug delivery, nanotechnology, alzheimer’s disease, Amyloid-β, solid lipid nanoparticles, SHSY5Y cells

## Abstract

Aggregation of Amyloid-β (Aβ) leads to the formation and deposition of neurofibrillary tangles and plaques which is the main pathological hallmark of Alzheimer’s disease (AD). The bioavailability of the drugs and their capability to cross the BBB plays a crucial role in the therapeutics of AD. The present study evaluates the Memantine Hydrochloride (MeHCl) and Tramiprosate (TMPS) loaded solid lipid nanoparticles (SLNs) for the clearance of Aβ on SHSY5Y cells in rat hippocampus. Molecular docking and in vitro Aβ fibrillation were used to ensure the binding of drugs to Aβ. The in vitro cell viability study showed that the M + T SLNs showed enhanced neuroprotection against SHSY5Y cells than the pure drugs (M + T PD) in presence of Aβ (80.35µM ± 0.455 µM) at a 3:1 molar ratio. The Box–Behnken Design (BBD) was employed to optimize the SLNs and the optimized M + T SLNs were further characterized by %drug entrapment efficiency (99.24 ± 3.24 of MeHCl and 89.99 ± 0.95 of TMPS), particle size (159.9 ± 0.569 nm), PDI (0.149 ± 0.08), Zeta potential (−6.4 ± 0.948 mV), Transmission Electron Microscopy (TEM), Atomic Force Microscopy (AFM) and in vitro drug release. The TEM & AFM analysis showed irregularly spherical morphology. In vitro release of SLNs was noted up to 48 h; whereas the pure drugs released completely within 3 hrs. M + T SLNs revealed an improved pharmacokinetic profile and a 4-fold increase in drug concentration in the brain when compared to the pure drug. Behavioral tests showed enhanced spatial memory and histological studies confirmed reduced Aβ plaques in rat hippocampus. Furthermore, the levels of Aβ decreased in AlCl_3_-induced AD. Thus, all these noted results established that the M + T SLNs provide enhanced neuroprotective effects when compared to pure and individual drugs and can be a promising therapeutic strategy for the management of AD.

## 1. Introduction

Alzheimer’s disease (AD) is a progressive neurodegenerative disorder whose pathology is mainly driven by the presence of plaques and neurofibrillary tangles containing Amyloid-β (Aβ) and hyper-phosphorylated tau protein [1]. According to recent WHO reports, AD is the seventh leading cause of death and more than 55 million people live with dementia worldwide. The number is expected to rise to 78 million in 2030 and 139 million in 2050 [2]. Aβ is a protein that is produced in normal physiology, and is involved in reducing the excitatory activity of potassium channels and reducing neuronal apoptosis [3]. Impaired Aβ clearance and mutations in the human APP gene cause the development of Aβ plaques and AD-like brain pathology [4]. Until June 2021 there was no particular drug targeting Aβ except a few symptomatic treatments such as acetylcholinesterase inhibitors & NMDA antagonists [5]. An Aβ antibody Aducanumab was recently approved by FDA which targets Aβ plaques the main hallmark [6]. The attributes that play a crucial role in delivering drugs to achieve a desired therapeutic outcome for treating AD are the blood brain barrier (BBB) penetration and bioavailability [7]. Nanotechnology-based delivery systems emerge as a ray of hope to overcome these drawbacks [8]. The lipid-based nanodelivery systems are promising tools for delivering drugs across BBB in comparison with conventional forms [9,10,11].

Solid lipid nanoparticles (SLN), among other nanoparticulate formulations, have lately received new consideration as a possible drug delivery system for brain targeting [12,13,14]. In order to treat neurodegenerative disorders in a nontoxic, safe, and efficient manner by crossing the BBB, solid lipid nanoparticles (SLNs) are one of the safest and least expensive drug carriers [12,13]. SLNs’ functionality and effectiveness depend on their constituents, size, shape, physico-chemical properties, and the synthetic processes by which they are formed [14,15,16]. Here we make use of the SLNs in encapsulating memantine Hydrochloride (MeHCl) and Tramiprosate (TMPS). MeHCl is an NMDA antagonist and is known to slow down the neurotoxicity and reduce Aβ levels involved in AD [17]. Tramiprosate is an orally administered compound that binds to various amino acid residues of Aβ_1-42_ [18] which results in the stabilization of Aβ monomers thereby preventing the formation of oligomers and fibrils.

This inhibition leads to neuroprotection by preventing subsequent deposition of Aβ. In order to improve its efficacy and reduce the adverse effects lipid-based nanoparticulate delivery would be feasible. The current study aims at targeting amyloid-β fibrillation [19] using lipid-based nanoparticles [20,21] carrying Aβ inhibitors (TMPS) [22] along with anti-glutaminergic drugs (MeHCl) which are proposed to be a very vital target in the management of Alzheimer’s disease.

## 2. Materials and Methods

### 2.1. Materials

Memantine Hydrochloride (MeHCl) was a generous gift sample from Strides Pharmasciences ltd., Bangalore, Tramiprosate (TMPS), Thioflavin-T, MTT, and Aluminium chloride was procured from Sigma Aldrich. Labrafil, labrasol and gelucire 43/04 was a gift sample from Gattefosse, Germany. Aβ_1-42_ was procured from Tocris Bioscience, UK. Fetal Bovine Serum, Penicillin-Streptomycin, and GlutaMAX TM were procured from Thermo Fisher Scientific, Pittsburgh, PA, USA. DMEM Hams F12 media, MEM Media, and Dialysis membrane with a molecular weight cut off of 12 kD were procured from Himedia Laboratories Pvt, Ltd., Mumbai.

### 2.2. Methods

#### 2.2.1. Molecular Docking Studies of TMPS and MeHCl

The docking is carried out using a CDOCKER algorithm from Discovery studio that is based on simulated annealing which is simulated using a CHARMm force field. The 3D structure of Aβ_1-42_ (2BEG) was downloaded from a protein data bank before docking and processed to remove side chains, loops, and conformers. Simultaneously the preparation of ligands was carried out to remove the duplicates and fix valences after which the protein and ligands were subjected to docking. The binding site of the protein was identified through a receptor cavity tool using a site search and flood-filling algorithm. The compounds memantine hydrochloride and tramiprosate were docked with the defined sphere site using random conformations with 1000 steps of dynamics to choose the best possible result for interaction analysis [22].

#### 2.2.2. In Vitro Aβ Fibrillation Studies

The Aβ fibrillation study was conducted using a thioflavin-T assay which uses Aβ_1-42_ human peptide at a concentration of 10 µM combined with or without various concentrations of Memantine Hydrochloride (500, 250, 125, 63, and 31.5 μM) and Tramiprosate (300, 150, 75, 37.5 and 18.75 μM). The Aβ_1-42_ solution was incubated with the drug solution for 48 h and then 20 µM Thioflavin-T was added. The fluorescence was measured at 450 nm for excitation and 485 nm for emission. Curcumin was used as a positive control and 0.1% DMSO was diluent for all compounds [23].

#### 2.2.3. Neuroprotective Effects of MeHCl & TMPS

##### Neuronal Cell Line Procurement and Maintenance

Human Neuronal cells SHSY5Y (ATCC^®^ No. CRL-2266) were obtained from American Type Culture Collection (ATCC, Rockville, MD, USA). The cells were maintained using the complete media of DMEM Hams F12: MEM media (1:1 ratio), 10% *v*/*v* FBS, 1% *v*/*v* Penicillin-Streptomycin, and glutaMAX TM solution at 37 ± 0.5 °C, with 5% CO_2_ in a sterile condition.

##### Cell Viability of Aβ_1-42_

In order to establish a model of Aβ-induced toxicity, SH-SY5Y cells were treated with different concentrations of Aβ_1-42_ (200, 100, 50, 25, 12.5, 6.25 & 3.125 µM) for 24 hr. We examined the cell viability using MTT assay and ThT assay, respectively.

##### In-Vitro Cytotoxicity Studies

The main aim of this study was to identify the better active compound which is having maximum inhibition of Aβ aggregation with less drug concentration against SHSY5Y cells. The SHSY5Y cells (10 × 10^3^ cells/well) were dispersed in 96 well plates (100 µL/well) of complete MEM media for 36 h to attain 80% confluence. The cells were treated with Aβ_1-42_ and with various concentrations of MeHCl and TMPS as mentioned below. Cisplatin (100 μM) was used as a positive control (PC) and blank SLNs were used as vehicle control (VC). The plates were incubated with 5% CO_2_ at 37 ± 0.5 °C for 24 h and then MTT solution was added and was incubated for 1hr. The formed formazone crystals were dissolved using DMSO after discarding the media and the optical density (OD) was measured using a microplate reader at 570nm [24,25]. The mean OD of each set of wells was calculated and the % of cell viability was calculated using Equation (1).
% Cell viability = Mean OD (test − blank)/mean OD (control − blank) × 100(1)

##### Simultaneous Combination Assay

Based on the individual Neuroprotective assay of drugs, we determined the combinational neuroprotective effects of MeHCl & TMPS against SHSY5Y cells by using simultaneous combination assay. SHSY5Y cells were seeded at a density of 10 × 10^3^ cells per well on 96-well plates, and after 36 h of incubation, the cells were treated simultaneously with Aβ_1-42_, MeHCl & TMPS for 24 h.

At constant ratios (MeHCl 3: TMPS 1 molar ratio), different drug doses were combined using the IC_50_ values calculated from the previous cytotoxicity studies. The SHSY5Y cells were treated with different doses of MeHCl (30, 15, 7.5, 3.75, 1.87, 0.93 & 0.46 μM) and TMPS (10, 5, 2.5, 1.25, 0.625, 0.312 & 0.156 μM) for simultaneous estimation [26].

##### Memantine Hydrochloride (MeHCl)

The primary stock solution of pure MeHCl (100 mM) was prepared using DMSO. Then, SHSY5Y cells were treated with various concentrations of pure MeHCl (3.9, 7.81, 15.62, 31.25, 62.5, 125, 250 μM).

##### Tramiprosate (TMPS)

The primary stock solution of pure TMPS, (100 mM) was prepared using DMSO. Then, SHSY5Y cells were treated with various concentrations of pure TMPS (7.81, 15.62, 31.25, 62.5, 125, 250, and 500 μM).

#### 2.2.4. Formulation of Solid Lipid Nanoparticles

SLNs were prepared by the homogenization-ultrasonication method [27]. The drug was dispersed in the lipid phase of labrasol and gelucire 43/04 (2:1) and heated to a temperature higher than the melting point of the lipid. A Smix of tween 80 and labrafil (3:1) finalized in the DoE was dissolved in the aqueous phase and heated to the same temperature as that of the lipid mixture. The aqueous phase was added into the lipid phase dropwise to form a primary emulsion. The formed primary emulsion was homogenized and probe sonicated for a specific time. The mixture was left to cool down to form solid lipid nanoparticles. Further, the formed nanoparticles were characterized for morphology and particle size.

#### 2.2.5. Experimental Design

In order to demonstrate the response of the surface model by attaining different combinations of values JMP pro software was employed. Without using the 3-level factorial design, we used the Box Behnken design to form a quadratic. A 3-factor, 3-level Box Behnken design was used to optimize the procedure for the formulation of SLNs. The selected independent variables are Smix, homogenization time, and homogenization speed, while the selected dependent variables were particle size and polydispersity index. This leads to the optimization of SLNs with a small experimental design (17 runs).

### 2.3. Evaluation of Drug Loaded SLNs

#### 2.3.1. %Drug Entrapment Efficiency (%DEE)

The entrapment efficiency of MeHCl and TMPS loaded and was determined by centrifuging a fixed amount of desired SLNs for 1 h at 10,000 RPM to obtain the supernatant. This supernatant was diluted further and the %DEE of both MeHCl and TMPS was determined using HPLC. The %DEE was calculated using Equation (2).
(2)$$\mathrm{Entrapment}\;\mathrm{Efficiency}=\left(\mathrm{Wd}-\frac{\mathrm{Ws}}{\mathrm{Wd}}\right)\times 100$$

#### 2.3.2. In Vitro Drug Release Studies

Release studies of pure drug memantine hydrochloride and tramiprosate pure drug (M + T PD) and memantine hydrochloride and tramiprosate loaded solid lipid nanoparticles (M + T SLNs) were performed by using a dialysis membrane to which a solution equivalent to 1mg/mL concentration of pure drug and drug loaded SLNs were loaded. The loaded dialysis bags were placed in 7.4 pH PBS at 37 ± 0.5 °C with 100 RPM stirring in an orbital shaker incubator. 1ml Aliquots were withdrawn and replaced at various time points (0.15, 0.5, 1, 1.5, 2, 2.5, 3, 4, 8, 12, 24, and 48 h). The concentration of MeHCl and TMPS in the sample aliquots was determined using the HPLC method.

#### 2.3.3. Transmission Electron Microscopy (TEM)

TEM was employed for morphological analysis and particle size confirmation. An arrangement of bright field imaging at collective magnification and diffraction approach was employed to disclose the form and size of the SLNs. The sample was prepared by diluting the optimized M + T SLNs in distilled water (1:100), and a sample was loaded on a copper grid and blemished with uranyl acetate for 30 s. The stained grid was dried, positioned on a glass slide and a coverslip, and observed under the microscope [28].

#### 2.3.4. Atomic Force Microscopy (AFM)

Atomic force microscopy (5600 LS, Agilent, Santa Clara, CA, USA) was used to perceive the surface morphology of M + T SLNs. The samples were dehydrated by placing them on silicon wafers at room temperature. The morphology of M + T SLNs was examined using contact scanning probe microscopy.

### 2.4. Animals

All experimental animals were acclimated to the laboratory environment for a week prior to the start of the experiment for in vivo pharmacokinetic, bio-distribution, and pharmacodynamics measurements. The Institutional Animal Ethics Committee (IAEC) granted approval for the submitted study procedure. The studies were carried out in accordance with the guidelines established by the institutional ethical committee of the JSS Academy of Higher Education and Research’s central animal facility.

### 2.5. Pharmacokinetics and Bio-Distribution

The 37 rats were divided into 2 groups, i.e., group I (M + T PD) and group II (M + T SLNs). Albino Wister Rats were injected intraperitoneally (i.p.) with 10 mg/kg of TMPS and 20 mg/kg of MeHCl in both pure drug and SLNs form. Following was the administration of both M + T.

PD and M + T SLNs the blood was withdrawn from the retro-orbital sinus at 0.15, 0.30, 1, 6, 12, 24, and 48 h. The obtained blood samples were centrifuged at 4000 RPM for the separation of plasma and stored at −80 °C until the analysis of MeHCl and TMPS content. All of the animals were anesthetized using ketamine before sacrificing. Brain tissues and other major organs, such as livers, kidneys, and spleens were excised, rinsed, and store at −80 °C to assess the bio-distribution of MeHCl and TMPS. The organs were homogenized individually using a tissue homogenizer and centrifuged at 12,000 RPM to separate the supernatant. The obtained supernatant was analyzed for the MeHCl and TMPS content using HPLC [29,30]. 

### 2.6. Pharmacodynamics

The animals were alienated into eight groups I–VIII. Group I was used as a control, whereas group II–VIII animals were administered with 100 mg/kg of AlCl_3_ orally for 4 weeks to provoke AD along with 0.9% of NaCl (5 mL/kg), AlCl_3_ + 10 mg/kg of MeHCl PD, AlCl_3_ + 20 mg/kg of TMPS PD, AlCl_3_ + 30 mg/kg of M + T PD, AlCl_3_ + 10 mg/kg of MeHCl SLN, AlCl_3_ + 20 mg/kg of TMPS SLN and M + T SLN single dose every day for 4 weeks respectively [31]. After the treatment period, the learning and memory of the animals were examined by the Morris Water Maze test. The brain tissues were excised, rinsed, and stored in neutral buffered formalin (NBF) which will further be used for histopathology studies. ELISA will be used for the quantification of Aβ_1-42_ in both control and treated groups using the commercial assay kits as per the protocols suggested by the manufacturer.

#### 2.6.1. Morris Water Maze Test

The strength of the learned spatial search bias was assessed during a probe trial on the 6th day without the platform. The MWM test was conducted as per the standard method by [32], with a minor modification to determine the impact of AlCl_3_-induced AD on spatial memory in Wistar rats. A circular drum (diameter: 125 cm; height: 36 cm) filled with water was split equally into four quadrants. Skimmed milk was used to make the water turbid. The platform was placed in the NW quadrant, 1 cm underneath the water. The rats were exposed to the acquisition trial (exercise to find the hidden platform) twice a day for five days. A probe test was carried out on the 6th day; the platform was removed to test the retention memory of the rats. The rats were allowed to swim in the drum for a period of 60 s, the assessment was video recorded and ANY-maze software was used to determine the escape latency, distance travelled, and the number of entries in the target quadrant and track plot of the mice.

#### 2.6.2. Histopathology

A 10% neutral buffered formalin (NBF) was used as a fixing solution for rat brains. Brain tissues were embedded in paraffin and coronal sections (3–5 μm) of the hippocampus region were cut using a microtome. Sections were mounted on a slide, washed and dehydrated with 95% ethanol, and stained with congo red dye for histopathological examination [33].

#### 2.6.3. ELISA

The Aβ content was measured using ELISA Kit following the kit’s protocol (Cloud Clone Corp., Katy, TX, USA).

## 3. Results

### 3.1. Docking Analysis

The hydrophobic interactions and the salt bridge between Asp23/Glu22 and Lys28 residue in Aβ_1–42_ is mainly responsible for the formation of insoluble aggregates and changes in β-sheet conformation. In order to demonstrate the binding modes of MeHCl and TMPS with Aβ_1-42_ we used the CDOCKER application where figures were generated through visualization. A 2BEG was a form Aβ which was chosen for the docking study of Aβ with MeHCl and TMPS. All docked conformations are ranked based on docking scores.

As shown in Figure 1, TMPS was mainly stabilized by conventional hydrogen bonding with Glu22 and Ala21. Furthermore, van der Waals interaction was seen with the Leu34, Val36, Ala21, and Glu22 and Carbon hydrogen bond interaction with Asp23, Glu22, and Leu34. However, there was a presence of an unfavorable bump in the interaction of MeHCl with Aβ_1-42_, thus proving that there is no role of memantine in the deaggregation of Aβ_1-42,_ but is made use of in the study due to its effect as an NMDA antagonist in the treatment of AD.

### 3.2. In Vitro Aβ Fibrillation Studies

Thioflavin-T is a dye that particularly binds to Aβ protein and helps in monitoring Aβ fibrillation. This study was used to evaluate the anti-amyloidogenic activity of MeHCl and TMPS. Both the drugs studied exhibited bioactivity. Tramiprosate showed higher bioactivity by inhibiting 16.56% of Aβ while Memantine Hydrochloride showed 3.22% inhibition of aggregation (Figure 2).

### 3.3. Neuroprotective Effects of MeHCl & TMPS

#### 3.3.1. Aβ_1-42_

The morphological changes and dose dependent responses of SHSY5Y on the treatment of Aβ_1-42_ are represented in Figure 3. As the dose increased (200, 100, 50, 25, 12.5, 6.25 & 3.125 µM) the aggregation around the cells also increased which was assessed using MTT & ThT assay. These results indicated that the cells were sensitive to Aβ_1-42_ exposure. The obtained IC 50 value for SHSY5Y cells with Aβ_1-42_ was 80.35 µM ± 0.455 µM.

#### 3.3.2. MeHCl

The morphological changes and dose dependent responses of SHSY5Y on the treatment of MeHCl was represented in Figure 4A. With the increase in the dose (3.9, 7.81, 15.62, 31.25, 62.5, 125, 250 μM) the aggregation around the cells also decreased. The obtained IC 50 value for SHSY5Y cells with MeHCl PD, MeHCl SLN, MeHCl PD + Aβ & MeHCl SLN + Aβ was 582.6 ± 2.098, 485.9 ± 4.196, 30.28 ± 4.196 & 10.67 ± 4.268 respectively (Figure 4).

#### 3.3.3. TMPS

The morphological changes and dose dependent responses of SHSY5Y in the treatment of TMPS are represented in Figure 4B. With the increase in the dose (7.81, 15.62, 31.25, 62.5, 125, 250, and 500 μM) the aggregation around the cells also decreased. The obtained IC 50 value for SHSY5Y cells with TMPS PD + Aβ & TMPS SLN + Aβ was 9.892 ± 1.56 & 9.535 ± 1.651 respectively (Figure 4).

#### 3.3.4. Simultaneous Estimation

Then, we examined the combinational neuroprotective effects of the MeHCl & TMPS combination against the SHSY5Y cell line (Figure 5). The cell viability effects of different drug combinations were assessed. As anticipated, the MeHCl & TMPS treatment exhibited the highest neuro-protective effect against SHSY5Y cells when used in combination. Ten µM of TMPS + 20 µM of MeHCl showed a cell viability of 89.159 ± 1.916 which is much higher than the individual cell viability. The IC50 of M + T PD and M + T SLN was found to be 5.235 ± 0.41 and 3.627 ± 0.56. This shows that the drugs used in combination show enhanced neuroprotection.

### 3.4. Formulation of Solid Lipid Nanoparticles

#### 3.4.1. Model Fitting

To fit the two target variables, statistical analysis and joint model fitting based on the actualized experimental design were carried out using JMP Pro^®^. The *p*-value was used to evaluate the significance of the overall effects and investigated interactions. After manual backward exclusion of the 95% confidence level, Table 1 displays the significant effects and interactions present in the model. From top to bottom, the associated *p*-values are decreasing. The impact of the associated effects or interactions on the responses increases with decreasing *p*-value. The “^” denotes the main effects with higher-level interactions (Table 1). Table 2 lists the results of an independent analysis of the data about the corresponding impact of each factor on each response.

#### 3.4.2. Effect of Independent Variables on Responses

The mean PDI of the prepared SLNs was in the range of 0.12 ± 0.002 to 0.455 ± 0.025, and the mean diameter of the prepared SLNs was in the range of 65.04 ± 0.515 to 490 ± 1.63. Analysis of variance (ANOVA) based on Fisher’s ratio (F-ratio) of 0.267 indicates that the model established for PDI and PS is statistically significant. The ratio of the model of sum squares to the overall sum of squares, or the value known as r-squared (r^2^), was calculated and found to be 0.90 for PS and 0.96 for PDI. This shows that the regression model accounts for 90% and 96% of the variation in the response for PS and PDI respectively. Figure 6 shows the significant correlation between the determined values and those anticipated by the PS & PDI fitted model. The limited range of points along the red line and inside the 95% confidence interval for model significance corroborates this (Figure 6).

#### 3.4.3. Verification of the Model

The formulation was replicated at the optimal conditions (15% Smix, 6 min of homogenization, and 15,000 RPM homogenization speed) predicted by the prediction profiler obtained from the box-Behnken design, thereby validating the desirability equations for response prediction (Figure 7). PS and PDI both had projected response values of 193.94 nm and 0.146, respectively. The experimental results of PS (159.90.081) and PDI (0.1540.0036) were in good agreement with the projected results. The experiment was performed in triplicate.

### 3.5. Evaluation of Drug Loaded SLNs

#### 3.5.1. %Drug Entrapment Efficiency (%DEE)

%DEE of MeHCl & TMPS was found to be 99.24 ± 3.24 and 89.99 ± 0.95, respectively.

#### 3.5.2. Determination of Particle Size (PS), Polydispersity Index (PDI), and Zeta Potential of Optimized Formulation

The particle size of the developed MeHCl + TMPS SLNs and Placebo formulations were found to be 159.9 ± 0.569 nm and 157 ± 0.623 nm with a PDI value of 0.149 ± 0.08 and 0.161 ± 0.04. The Zeta potential of the MeHCl + TMPS SLNs and Placebo formulations were found to be −6.4 ± 0.948 mV and −6.15 ± 0.854 mV.

#### 3.5.3. Transmission Electron Microscopy (TEM)

The morphological analysis of optimized M + T SLNs was examined using TEM micrographs. The particles were spherical with sizes varying in 0.5μm scale. The obtained data were in good agreement with the particle size analysis of the M + T SLNs formulation (Figure 8C).

#### 3.5.4. In-Vitro Cumulative % Drug Release Study

As shown in Figure 8A, the In-vitro % drug release study was performed with pure drug mixture solutions and M + T SLNs formulation. The pure drug released 100 ± 6% of MeHCl and 100 ± 7% of TMPS within 3 h, but the M + T SLNs released 91.73 ± 1.5% of MeHCl and 95.90 ± 1.51% of TMPS. About 80% of MeHCl and TMPS were released from SLNs slowly up to 48 h.

#### 3.5.5. AFM

AFM topography image (Figure 8B) further confirmed the particle size of SLNs in the range of 156nm. The morphology of the SLNs as per the AFM topography was irregularly spherical with a smooth surface. The different shapes of the nanoparticles may be attributed to the mechanism of formation of the nanoparticles.

#### 3.5.6. Stability Studies

The effect of different storage conditions on the particle size and PDI of MeHCl + TMPS SLNs. MeHCl + TMPS SLNs stored at 4 ± 2 °C showed a slight increase in particle from 159.9 ± 0.0 nm to 162.35 ± 0.0707 nm, whereas the PDI also increased slightly from 0.154 ± 0.0 to 0.182 ± 0.0007 from day 1 to day 90, respectively.

Similarly, the storage condition of 25 ± 2 °C showed a slight but lesser increase in particle size when compared to 4 ± 2 °C. The particle size was 159.9 ± 0.0 nm on day 1 which increased gradually to 160.45 ± 0.070 nm on day 90. Furthermore, the PDI increased from 0.154 ± 0.0 to 0.161 ± 0.0007 on day 1 and day 90, respectively. The stability data revealed no significant changes in particle size and PDI in both 4 ± 2 °C and 25 ± 2 °C, which shows that the formulation is stable at both storage conditions (Figure 9).

### 3.6. Pharmacokinetics

The results of the pharmacokinetic parameters with an i.p. injection of M + T PD and M + T SLNs are represented in Table 3 and Figure 10. While the M + T PD solution demonstrated a quick initial clearance rate from the blood within 1 h of administration, M + T SLNs still showed delayed blood clearance rates even after 4 h of administration. In comparison to M + T SLNs, the significant pharmacokinetics characteristics of M + T PD showed a shorter mean residence time (MRT). The peak plasma concentration (C_max_) was found to be 144.601 ± 0.354 & 57.018 ± 0.2029 for M + T PD and 204.79 ± 0.042 and 65.618 ± 0.292 for M + T SLNs. The increase in the AUC_0-ꚙ_ in M + T SLNs might be due to the avoidance of first-pass metabolism by lymphatic transport. Additionally, M + T SLNs had considerably lower plasma clearance than M + T PD (Table 3). Figure 10 shows the Pharmacokinetic graph of M + T PD & M + T SLNs administered intraperitoneally at 30 mg/kg body weight (1:3 molar ratio). These results suggest that M + T SLNs have improved pharmacokinetic profiles when compared to the pure drug.

### 3.7. Bio Distribution

The highest concentration was found in the brain after dosing with M + T SLNs i.e., 177.9598 ± 18.366291 & 30.29417 ± 2.012082 µg/mL of MeHCl & TMPS, respectively, which (Figure 11) might be because of the faster and better absorption by the brain. The concentration of both drugs was very minimal in the liver (62.35548836 ± 13.335808 & 13.79340481 ± 3.012082 µg/mL), spleen (36.345 ± 9.169307 & 3.621701 ± 1.912082 µg/mL) and kidneys (18.96022784 ± 12.036123 & 2.26547381 ± 1.012082) of MeHCl & TMPS respectively in comparison to that of the pure drugs.

M + T PD, when delivered intraperitoneally, demonstrated considerably lower brain concentrations (48.07 ± 3.050 & 5.015 ± 7.031 µg/mL) than the drug-loaded SLNs. The pure drug concentration was very high in the liver (174.929 ± 3.505 & 31.650 ± 7.012 µg/mL) followed by other organs such as the spleen (61.239 ± 6.067 & 3.621 ± 2.012 µg/mL) and kidneys (50.467 ± 7.828 &2.717 ± 2.012 µg/mL).

### 3.8. Pharmacodynamics

#### 3.8.1. Morris Water Maze (MWM)

The impact of AlCl_3_-induced AD in rats on hippocampal-dependent spatial memory was assessed by the MWM test. As shown in Figure 12E, AlCl_3_-treated rats showed increased latency time to find the hidden platform (*p* < 0.01) when compared with normal rats. AlCl_3_-treated rats spent significantly less time (*p* < 0.05), lesser number of entries (*p* < 0.01) and covered less distance (*p* < 0.01) in the target quadrant compared when compared with normal rats. This indicates that AlCl_3_ successfully induces AD in rats. M + T SLNs (10 + 5 mg/kg) treated rats showed reduced latency time to find the hidden platform, increased number of entries in the target quadrant, spend more time, and covered more distance in the target quadrant (Figure 12A–E). These data indicate that M + T SLNs restores spatial memory.

#### 3.8.2. ELISA

We performed an ELISA assay to determine the impact of AlCl_3_, MeHCl & TMPS pure drug and M + T PD, MeHCl & TMPS SLNs, and M + T SLNs on the levels of Aβ in the hippocampal region of rat brain. Normal Rat hippocampus showed the lowest concentration of Aβ whereas the AlCl_3_ treated with the highest concentration of Aβ. There was a significant decrease in the levels of Aβ (*p* < 0.01) in M + T SLNs when compared to that of M + T PD upon simultaneous induction of AD using AlCl_3_ and treatment with the drugs (Figure 13).

#### 3.8.3. Histopathology

Microscopic examination of CA1, CA2, and DG region of the hippocampus using congo red stain was carried out to examine the deposition of Aβ in various treatment groups. AlCl_3_-induced AD showed marked deposition of Aβ in CA1, CA2, and DG regions of rat hippocampus, whereas the control group showed normal hippocampal neurons with no deposition of Aβ. Among the treatment groups, the SLN-treated groups showed the deposition of very few foci of Aβ in various regions of the hippocampus when compared to that of the pure drug groups which showed deposition of multiple foci of Aβ (Figure 14).

## 4. Discussion

The current study’s objective was to develop and characterize solid lipid nanoparticles for the elimination of Aβ in Alzheimer’s disease using the homogenization-ultrasonication technique. The use of SLNs can increase bioavailability without the use of high doses by passing physiological barriers, guiding the active compound towards the target site with a significant reduction in toxicity for the surrounding tissues, and protecting drugs from chemical and enzymatic degradation [34]. Memantine Hydrochloride (MeHCl) and tramiprosate (TMPS) were the drugs chosen to be administered in combination as both of them have shown potential in inhibiting Aβ aggregation. Previous research suggests that Memantine hydrochloride an NMDA antagonist has Aβ disaggregation activity [35] but showed no interaction with Aβ (2BEG) in molecular docking studies, however, in contrast, it showed 3.22% inhibition of aggregation in thioflavin-T-based in vitro assay. Tramiprosate showed good interaction with various amino acid residues of the Aβ protein [36] and inhibited 16.56% of Aβ aggregation thus proving its potential in Aβ sheet disaggregation.

The in vitro % cell viability studies of SHSY5Y cells in the treatment of Aβ, MeHCl PD, and SLNs, and TMPS PD and SLNs were evaluated by MTT assay. It was seen that as the concentration of Aβ increased the cell viability decreased. The IC_50_ value for Aβ was 80.35µM ± 0.455 µM which was used for further cell viability studies. In the case of MeHCl PD and TMPS PD, as the concentration increased the cell viability decreased which may be attributed to the toxicity of the pure drug. In presence of Aβ, the cell viability increased as the concentration increased which suggests that the treatment is reducing the Aβ concentration with an IC_50_ of 30.28 ± 4.196 for MeHCl PD + Aβ & 9.89 ± 1.56 for TMPS PD + Aβ. The drugs, when formulated into SLNs, showed an increase in cell viability with an increase in concentration up to 88% for MeHCl SLN + Aβ and 97% for TMPS SLN + Aβ. It can also be seen that the IC_50_ of MeHCl SLN + Aβ was 10.67 ± 4.268 and TMPS SLN + Aβ was 9.535 ± 1.651 which was the lowest IC_50_ when compared to that of the pure drug. Taking into consideration the obtained results, we conducted a simultaneous combination assay at a 3:1 molar ratio, which resulted in enhanced neuroprotection when compared to that of individual pure drug and SLNs, and also resulted in a much lower IC_50_; 5.235 ± 0.41 for M + T PD and 3.627 ± 0.56 for M + T SLN. Hence, we considered the 3:1 molar ratio for further studies.

The M + T SLNs were prepared using the homogenization-ultrasonication method using Labrasol as liquid lipid and gelucire 43/04 (2:1) as solid lipid which was chosen based on the drug solubility in the respective lipids. The surfactants used were tween 80 and co-surfactant labrafil (3:1) was used which was selected by constructing ternary phase diagrams. The Box-behnken design was used to optimize M + T SLNs where homogenization time, homogenization speed, and smix ratio were independent variables and Particle size and PDI were dependent variables. The predicted response values for PS and PDI were 193.94nm and 0.146 respectively which is well in agreement with the experimental values having PS of 159.9 ± 0.081 and PDI of 0.154 ± 0.0036.

The particle size of less than 200nm will effectively cross the blood brain barrier [37] and lower the polydispersity index more uniform the particle distribution. A negative zeta potential of -6.4mV shows that the particles are neutral [38]. The % of entrapment efficiency ensures that the chosen lipids efficiently entrap the drugs. The spherical morphology was confirmed by both TEM and AFM. The in vitro drug release studies showed that the pure drugs released completely within 3hrs whereas the M + T SLNs showed a sustained release up to 48hrs which followed the Higuchi model for drug release. The improved formulation’s excellent stability suggests that it can preserve integrity even when it is diluted in the body. The in-vitro characterization study was similar to earlier studies.

The in vivo pharmacokinetics and bio-distribution study in rats following a single i.p. dose was investigated using the HPLC technique. The maximum plasma concentration (C_max_) and the time required to reach it (T_max_) were directly calculated from the Plasma concentration-time profile. The non-compartmental model’s calculation of the additional crucial pharmacokinetic parameters was carried out utilizing Phoenix winnonlin software. The M + T SLNs showed improved pharmacokinetic parameters when compared to the pure drug (Table 3), thus proving the efficiency of the formulation. Closely, there was a 4-fold increase in the concentration of SLNs in the brain compared with pure drug-treated brains in rats. The biodistribution pattern strongly suggests that MeHCl + TMPS is present in the lipid carrier in intact form, allowing the medication to pass the blood-brain barrier.

The pharmacodynamics study was carried out on AlCl_3_-induced AD in rats. The effect of AlCl_3_ on spatial memory was assessed using Morris water maze studies which showed that the M + T SLNs enhanced spatial memory. This was proven by the increased number of entries to the target quadrant, increased time spent, distance travelled, and decreased escape latency in M + T SLN treated group when compared with the control group. ELISA assay was used to quantify the Aβ protein in rat hippocampus in various treatment groups. Reduced concentration of Aβ in the M + T SLN treated group shows that the treatment is effective in reducing the Aβ concentration in rat hippocampus. The histopathological studies using congo red dye also confirmed the presence of Aβ plaques in the AlCl_3_-induced group which decreased significantly in the M + T SLN-treated group.

All the in vitro and in vivo data confirmed that the M + T SLNs was superior when compared to that of pure drugs. This study also proved that the combinational therapy achieved the desired reduction of Aβ protein burden when compared to the individual pure drug and formulation. The currently available treatments do not focus on the pathological hallmarks of AD or reduce the protein burden in the brain [39]. The BBB would be one more major drawback in treating neuronal disorders which can be overcome by the use of lipid-based NPs [40,41]. The results of bio-distribution suggest that the M + T SLNs are capable of crossing the BBB. Based on the positive outcomes of the various in vitro characterizations of SLNs, we draw the conclusion that M + T SLNs can serve as an effective drug delivery system to cross BBB and manage AD. Treatments mainly targeting the patient’s pathological hallmark burden would help in better management of the disease [42].

## 5. Summary and Conclusions

In the current investigation, homogenization-ultrasonication was used to prepare solid lipid nanoparticles (SLNs), which were then optimized utilizing the box-Behnken design approach in JMP pro software. The effect of the independent factors on the PS and PDI of M + T SLNs was effectively examined. The TEM and AFM results showed the SLNs being roughly spherical and having maximum entrapment efficiency which was stable over 60 days. Both medications were released steadily from SLNs, according to in vitro drug release experiments. Furthermore, the increased safety and effectiveness of M + T SLNs have been verified by in vitro cell-based assays. Based on the constructive results obtained from the various in vitro characterizations of SLNs we conclude that the M + T SLNs can be a useful carrier to deliver drugs across BBB.

## Figures and Tables

**Figure 1 pharmaceutics-15-00221-f001:**
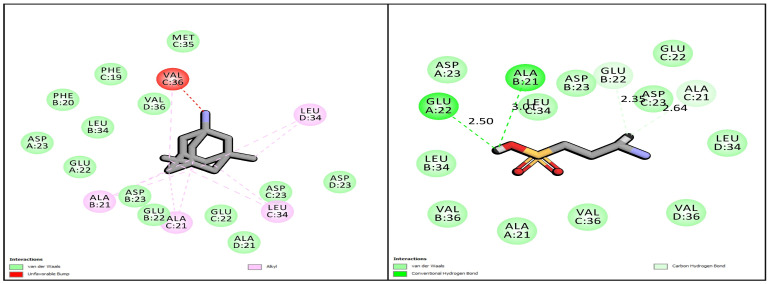
2D molecular docking simulations of MeHCl and TMPS.

**Figure 2 pharmaceutics-15-00221-f002:**
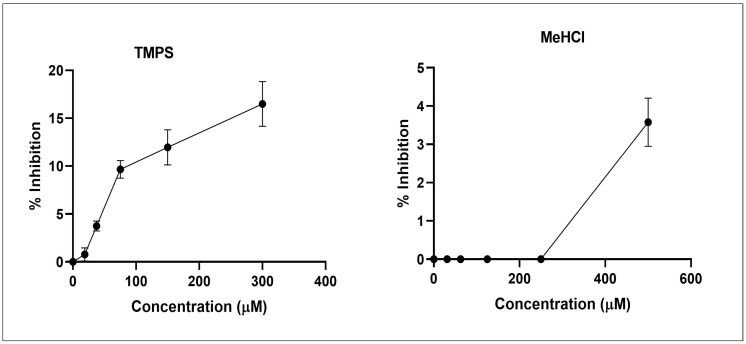
Amyloid-β anti-aggregation assay (data points represented as mean ± SD, where *n* = 3 and *p* < 0.05).

**Figure 3 pharmaceutics-15-00221-f003:**
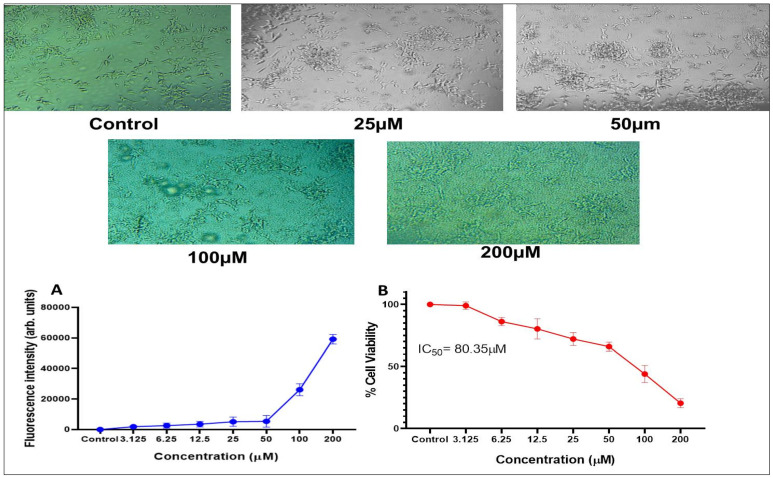
Effect of different Aβ_1-42_ concentrations on the viability of SHSY5Y cells (**A**) ThT assay (**B**) MTT assay (data points mean ± SD, *n* = 3).

**Figure 4 pharmaceutics-15-00221-f004:**
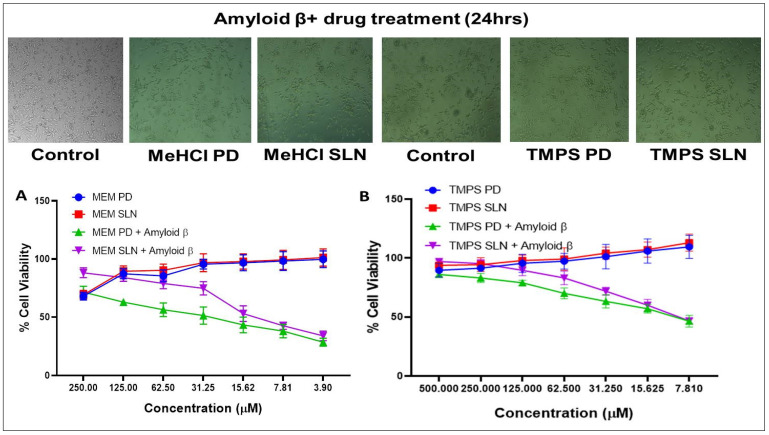
The % of Cell viability assay. (**A**) The dose dependent response of SHSY5Y cells on the treatment of MeHCl (**B**) The dose dependent response of SHSY5Y cells on the treatment of TMPS. Data is represented in mean ± SD (*n* = 3).

**Figure 5 pharmaceutics-15-00221-f005:**
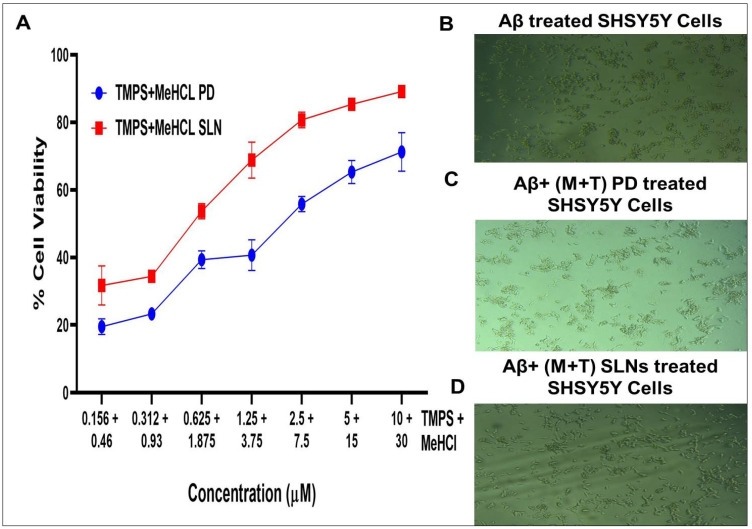
(**A**) The dose dependent activity of MeHCl & TMPS combination on SHSY5Y cells, (**B**) Aβ treated SHSY5Y cells, (**C**) Aβ+ (M+T) PD treated SHSY5Y cells and (**D**) Aβ+ (M+T) SLNs treated SHSY5Y cells. Data is represented in mean ± SD (*n* = 3).

**Figure 6 pharmaceutics-15-00221-f006:**
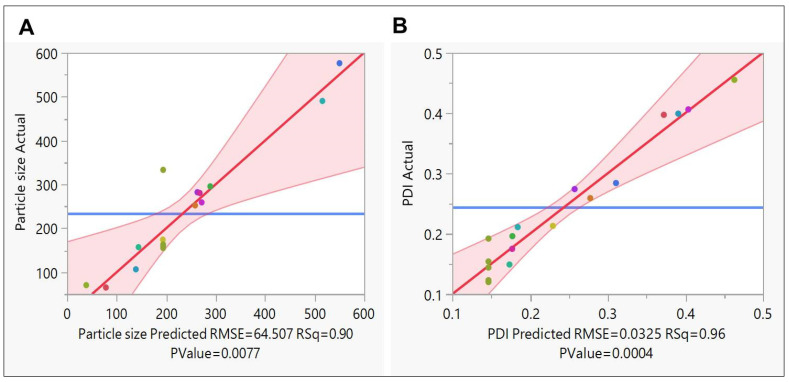
Statistical evaluation of prepared SLNs of (**A**) Predicted v/s actual plot for Particle size and (**B**) Predicted v/s actual plot for PDI.

**Figure 7 pharmaceutics-15-00221-f007:**
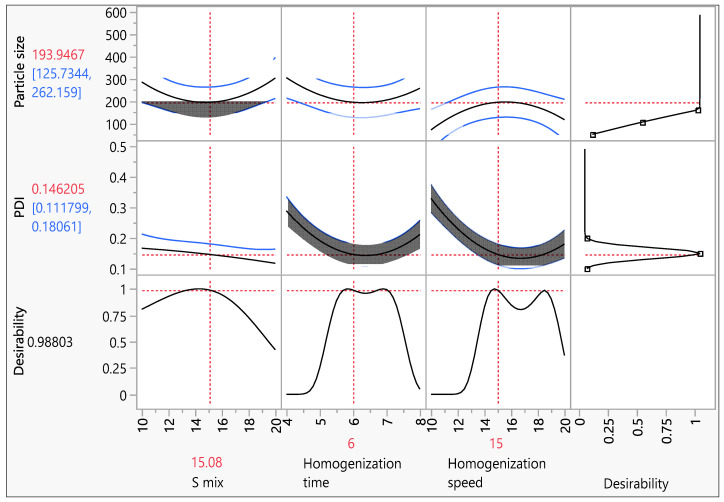
Prediction profile of prepared SLNs where the actual values are similar to that of the predicted values.

**Figure 8 pharmaceutics-15-00221-f008:**
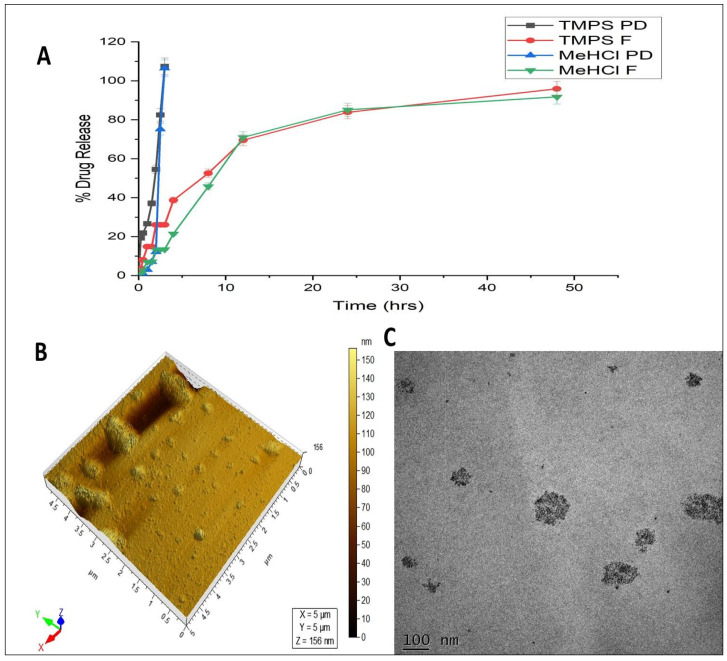
(**A**) Drug release profile of MeHCl & TMPS in pure form and SLNs where the data is represented in mean ± SD (*n* = 3) (**B**) AFM image of M + T SLN (**C**) TEM micrographs of M + T loaded SLNs.

**Figure 9 pharmaceutics-15-00221-f009:**
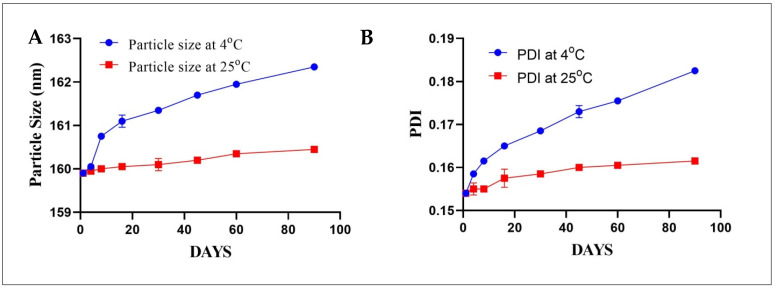
Effect of storage condition on particle size and PDI of M + T SLNs (**A**) Particle Size (**B**) PDI. The data is presented in mean ± SD (*n* = 3).

**Figure 10 pharmaceutics-15-00221-f010:**
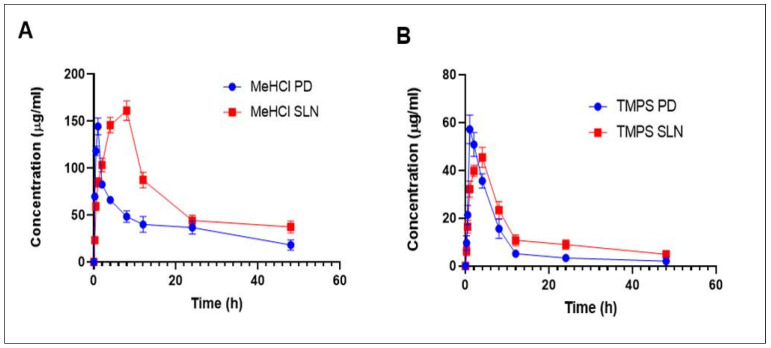
Plasma concentration-time profile of MeHCl + TMPS PD and MeHCl + TMPS SLNs administered intraperitoneally at 30 mg/kg (1:3 molar ratio) (**A**) Plasma concentration-time profile of MeHCl PD & MeHCl SLN (**B**) Plasma concentration-time profile of TMPS PD & TMPS SLN. The data is represented in mean ± SD (*n* = 3).

**Figure 11 pharmaceutics-15-00221-f011:**
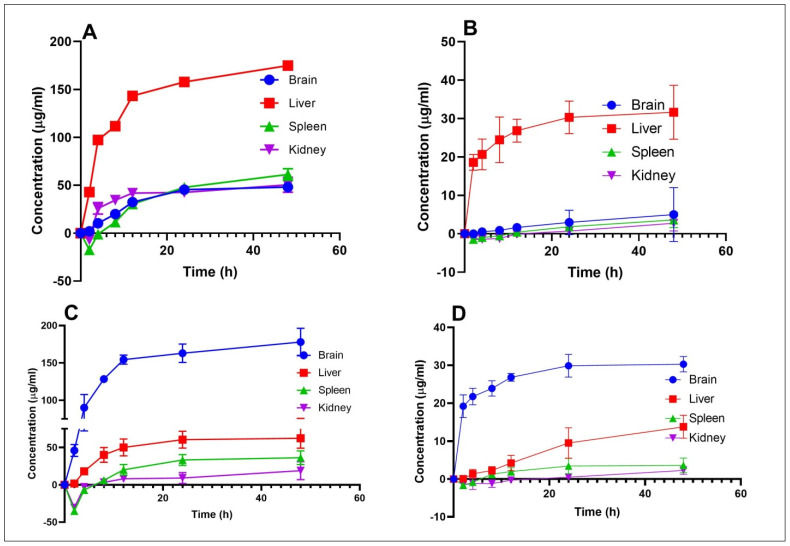
MeHCl and TMPS Concentration in major organs after intraperitoneal administration at 30mg/kg body weight (1:3 molar ratio) (**A**) MeHCl PD (**B**) TMPS PD (**C**) MeHCl SLN (**D**) TMPS SLN. Data are represented in mean ± SD (*n* = 3).

**Figure 12 pharmaceutics-15-00221-f012:**
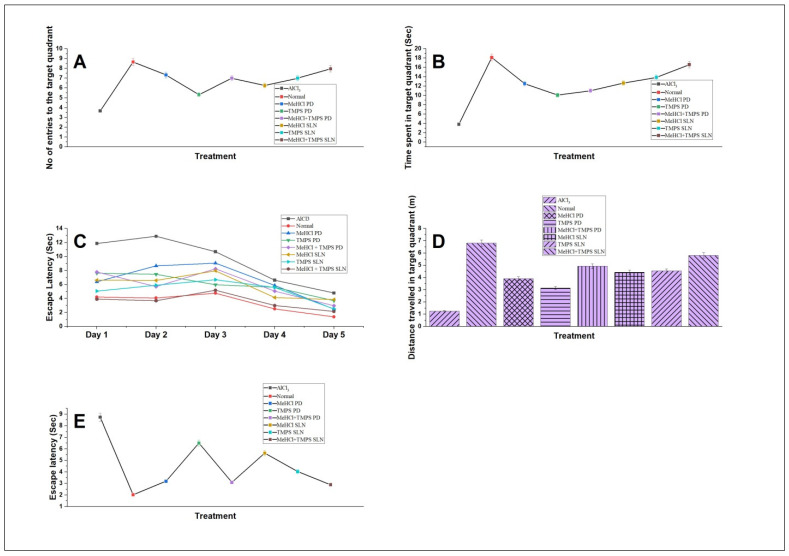
Daily administration of MeHCl & TMPS improved spatial memory in rats which was demonstrated by the (**A**) No. of entries in the target quadrant (**B**) Time spent in the target quadrant (**C**) Learning pattern in the training phase (**D**) Distance travelled in the target quadrant (**E**) Escape latency where all the results showed that AlCl_3_ induced AD and M + T SLNs showed the best results in comparison with other treatment groups. All the data is represented in mean ± SD (*n* = 3) with *p* < 0.01.

**Figure 13 pharmaceutics-15-00221-f013:**
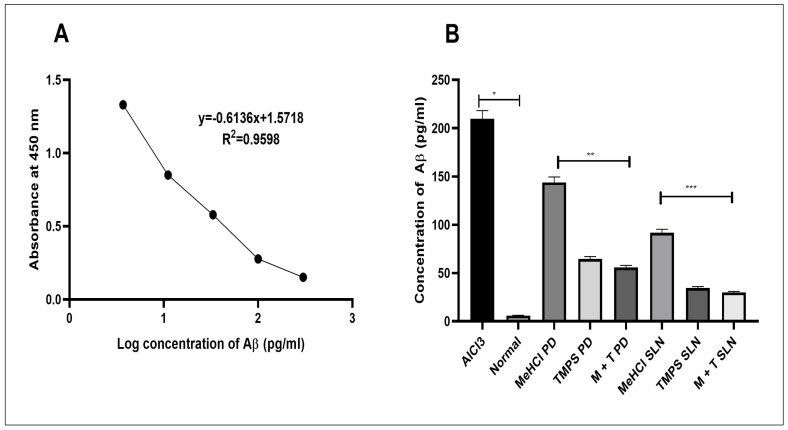
Determination of Aβ concentration in AlCl_3_ induced AD in Rat brain (**A**) Aβ standard curve (**B**) Reduced Aβ concentration in MeHCl + TMPS treated rat brain with data represented as mean ± SD (*n* = 3). * indicates *p* < 0.01, ** indicates *p* < 0.05 and *** indicates *p* < 0.001.

**Figure 14 pharmaceutics-15-00221-f014:**
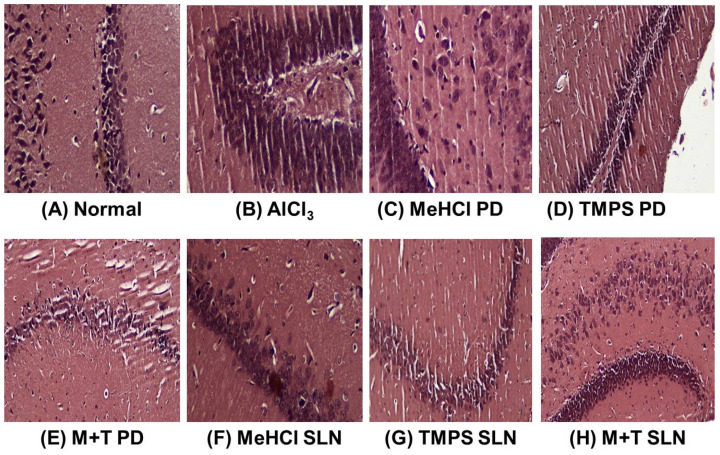
Histopathology of Hippocampal region of rat brain stained with congo red dye (**A**) Control group shows hippocampal neurons appeared normal without any deposition of Aβ (**B**) AlCl_3_ represents the deposition of foci of Aβ in the CA1, CA2, DG region of hippocampus also adjacent ventricle is filled with Aβ (**C**) MeHCl PD shows birefringence deposition of multiple foci of Aβ in CA2 region of the hippocampus (**D**) TMPS PD shows deposition of few foci of Aβ in CA1 region of the hippocampus (**E**) M + T PD shows deposition of few foci of Aβ in CA1 region of the hippocampus (**F**) MeHCl SLN showed deposition of few foci of Aβ in hippocampus neurons of CA1 region (**G**) TMPS SLN showed very few depositions of Aβ in CA1 region of the hippocampus (**H**) M + T SLN showed deposition of very few foci of Aβ deposition in various regions of the hippocampus.

**Table 1 pharmaceutics-15-00221-t001:** Effect Summary of factors and observed responses for MeHCl + TMPS SLNs.

Source	Log Worth	*p* Value	
Homogenization speed*Homogenization speed	3.612	0.00024	
Homogenization time*Homogenization time	3.492	0.00032	
Homogenization speed (10,20)	3.475	0.00033	^
Smix*Homogenization time	2.669	0.00214	
Homogenization time*homogenization speed	2.3922	0.00406	
Homogenization time (4,8)	1.916	0.01212	^
Smix*Smix	1.811	0.01545	
Smix*Homogenization speed	1.401	0.03970	
Smix (10,20)	1.158	0.06957	^

(^ denotes effects with containing effects above them).

**Table 2 pharmaceutics-15-00221-t002:** Box Behnken design factors and observed responses for MeHCl and TMPS loaded SLNs (* Mean ± SD, *n* = 3).

Pattern	S_mix_	Homogenization Time	Homogenization Speed	Particle Size * (Mean ± SD)	PDI * (Mean ± SD)
0	15	6	15	159.9 ± 0.569	0.154 ± 0.04
0	15	6	15	163 ± 0.070	0.192 ± 0.0007
+0−	20	6	10	252 ± 0.212	0.259 ± 0.0007
0−+	15	4	20	106.5 ± 0.424	0.399 ± 0.0007
−−0	10	4	15	576 ± 0.572	0.284 ± 0.0014
−0−	10	6	10	65.04 ± 0.282	0.397 ± 0.0007
0	15	6	15	333 ± 0.424	0.123 ± 0.0007
+0+	20	6	20	157 ± 0.282	0.149 ± 0.0007
+−0	20	4	15	282.1 ± 1.414	0.274 ± 0.0028
−0+	10	6	20	295.4 ± 0.353	0.196 ± 0.0014
0	15	6	15	158 ± 0.282	0.12 ± 0.0028
0	20	8	15	490 ± 0.353	0.211 ± 0.0028
−+0	10	8	15	174.1 ± 0.353	0.213 ± 0.0014
0+−	15	8	10	70.43 ± 0.296	0.455 ± 0.0028
0	15	6	15	155 ± 0.424	0.144 ± 0.0028
0++	15	8	20	280.4 ± 0.141	0.175 ± 0.0028

* *n* = 3.

**Table 3 pharmaceutics-15-00221-t003:** Pharmacokinetic parameters of MeHCl + TMPS PD & MeHCl + TMPS SLN were administered intraperitoneally at 30 mg/kg (1:3 molar ratio).

	M + T PD		M + T SLN	
Parameters	MeHCl PD *	TMPS PD *	MeHCl SLN *	TMPS SLN *
C_max_	144.601 ± 0.354	57.018 ± 0.2029	204.79 ± 0.042	65.618 ± 0.292
T_max_	1 ± 0	1 ± 0	4 ± 0	4 ± 0
Cl	7.509 ± 0.099	20.239 ± 0.1166	4.465 ± 0.134	12.05 ± 0.113
MRT	16.634 ± 0.475	10.437 ± 0.236	18.31 ± 0.241	15.22 ± 0.229
AUC_0-ꚙ_	2635.268 ± 0.118	491.537 ± 0.731	4573.705 ± 0.12	835.45 ± 0.478
AUC_0-48_	1854.468 ± 0.103	412.285 ± 0.502	3401.657 ± 0.289	614.376 ± 0.288
V_z_	325.604 ± 0.113	798.56 ± 0.602	136.938 ± 0.101	538.887 ± 0.229

* Mean ± SD (*n* = 3).

## Data Availability

Data will be made available on request to the corresponding author.

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
