# Peer review of "Development and Evaluation of Solid Lipid Nanoparticles for the Clearance of Aβ in Alzheimer’s Disease"

_pharmaceutics, 2023, doi:10.3390/pharmaceutics15010221_

Round 1
Reviewer 1 Report
The present research article by Goravinahalli Shivananjegowda and colleagues, entitled ‘Development and Evaluation of Solid Lipid Nanoparticles for the Clearance of Aβ in Human Neuroblastoma Cells and Rat Hippocampus’ is a well-written and useful summary on the status of knowledge of the importance of targeting amyloid-β fibrillation using lipid based nanoparticles carrying Aβ inhibitors in the management of Alzheimer’s disease.
The main strength of this manuscript is that it addresses an interesting and timely question, describing the use of solid lipid nanoparticles for the elimination of Aβ in Alzheimer's disease using the homogenization-ultrasonication technique. In general, I think the idea of this article is really interesting and the authors’ fascinating observations on this timely topic may be of interest to the readers of Pharmaceutics. However, some comments, as well as some crucial evidence that should be included to support the author’s argumentation, needed to be addressed to improve the quality of the manuscript, its adequacy, and its readability prior to the publication in the present form, in particular reshaping parts of the Introduction and Methods sections by adding more evidence and theoretical constructs.
Please consider the following comments:
· I suggest changing the title. In my opinion, in the present form it seems to be too wordy and not enough informative and appropriate.
· Abstract: According to the Journal’s guidelines, the abstract should be a total of about 200 words maximum. Please correct the actual one.
· A graphical abstract that will visually summarize the main findings of the manuscript is highly recommended.
· In general, I recommend authors to use more evidence to back their claims, especially in the Introduction of the article, which I believe is currently lacking. Thus, I recommend the authors to attempt to deepen the subject of their manuscript, as the bibliography is too concise: nonetheless, in my opinion, less than 50 articles for a research article are really insufficient. Therefore, I suggest the authors to focus their efforts on researching more relevant literature: I believe that adding more studies and reviews will help them to provide better and more accurate background to this study.
- Introduction: The ‘Introduction’ section is well-written and nicely presented, with a good balance of descriptive text and information about pathologic substrates of Alzheimer’s disorder. Nevertheless, I believe that more information about pathophysiology and core features of these disorders will provide a better and more accurate background, because as it stands, this information is not highlighted in the text. In this regard, I would suggest to add more information on pathological neural substrates of neurodegeneration in AD, specifically on structural as well as functional abnormalities of specific brain regions (i.e., hippocampus and prefrontal cortex), and on related and on related effects on patients’ cognitive impairments. In my opinion, authors could further explore significant structural brain alterations and impaired brain circuits in AD (https://doi.org/10.1016/j.tins.2022.04.003; DOI:10.1038/s41598-018-31000-9), and focus on relationship between the molecular regulation of higher-order neural circuits and neuropathological alterations in this neurodegenerative disorder (https://doi.org/10.3390/cells11162607; https://doi.org/10.3390/biomedicines9050517).
· Evaluation of Drug loaded SLNs: Please rewrite this paragraph and provide more detailed information about methods utilized to measure effect of drug release and its distribution in the organism.
· Results: In my opinion, this section is well organized, but it illustrates findings in an excessively broad way, without really providing full statistical details, to ensure in-depth understanding and replicability of the findings. I suggest rewriting this section more accurately, and to present statistical data not only in the main text, but also in tables.
· Discussion: In this final section, authors described the results of their research and their argumentation and captured the state of the art well; however, I would have liked to see some views on a way forward. I believe that the authors should make an effort, trying to explain the theoretical implication as well as the translational application of this paper, to adequately convey what they believe is the take-home message of their study. In this regard, I believe that it would be necessary to discuss theoretical and methodological avenues in need of refinement, as well as suggestions of a path forward in understanding the possibility of new treatments for AD.
· In according to the previous comment, I would ask the authors to also include a proper ‘Limitations and future directions’ section before the end of the manuscript, in which authors can describe in detail and report all the technical issues brought to the surface.
· Figures and Tables: According to the Journal’s guidelines, please add an explanatory caption for each figure within the text.
· References: Authors should consider revising the bibliography, as there are several incorrect citations. Indeed, according to the Journal’s guidelines, they should provide the abbreviated journal name in italics, the year of publication in bold, the volume number in italics for all the references. Also, please correct in-text citations: reference should be numbered, and placed in square brackets [ ] (for example [1]).
· Overall, I suggest submitting your work to an English native speaker to help with some grammar mistakes that can be found in different sections of the manuscript.
Overall, the manuscript contains 14 figures, 4 tables and 31 references. I believe that the manuscript might carry important value in describing the utilization of solid lipid nanoparticles for the elimination of Aβ in Alzheimer's disease using the homogenization-ultrasonication technique.
I hope that, after these careful revisions, this paper can meet the Journal’s high standards for publication.
I am available for a new round of revision of this paper. I declare no conflict of interest regarding this manuscript.
Best regards,
Reviewer
Reviewer 2 Report
The Manuscript: „ Development and Evaluation of Solid Lipid Nanoparticles for the Clearance of Aβ in Human Neuroblastoma Cells and Rat Hippocampus’’ by Meghana Goravinahalli Shivananjegowda and colleagues reports on development and characterization of solid lipid nanoparticles (SLN) for the elimination of Aβ in Alzheimer's disease using the homogenization ultrasonication technique. SLNs has been known to very good medium to supply central nervous system (CNS) drugs across the blood-brain barrier (BBB) in developing therapeutic strategy against CNS disorders such as Alzheimer’s disease (AD), Parkinson’s disease (PD), Huntington’s disease (HD), etc. The submitted study contributes further in bioavailability of SLNs in drug transfer beyond physical barriers in prophylaxis of AD. Thes tudy is nicely conducted with adequate description of methodology and with sufficient results. After going through the manuscript, I have few comments for the authors:
1. The abstract is messy with a lot of statistical data. Please refraining detailed statistical information in the abstract and make it more composed.
2. Please briefly mention the aim of your study in the last paragraph of Introduction section.
3. The information on experimental animals is lacking. Please include it under a separate subsection Animals.
Reviewer 3 Report
In this work the development and evaluation of solid lipid nanoparticles for the clearance of Aβ in human neuroblastoma cells and rat hippocampus are described. Authors used homogenization-ultrasonication to prepare solid lipid nanoparticles, which were then optimised utilising the box-Behnken design approach in JMP pro software. The work is of interest because based on the constructive results the M+T SLNs can be a useful carrier to deliver drugs across BBB. Taking into account the mentioned below notes I think that this article needs minor revision.
Notes:
1. There are some printings mistakes, which should be checked and corrected in the whole article. For example, “Nano technology” in the line 63 should be written together as “Nanotechnology”. At the line 87 before the word “Methods” the point should be deleted. In the line 254 “van der wals interaction” should be changed by “van der Waals interaction”.
2. The meaning of “M+T” and “BBB” abbreviations should be added in the text at the first mentioning.
3. Introduction should be complemented by the application and more information about solid lipid nanoparticles.
4. The information which is written in the insert area of Figure 12 should be increased, because it is not clear presented.
Round 2
Reviewer 1 Report
The authors did an excellent job clarifying all the questions I have raised in my previous round of review. Overall, this is a timely and needed work. It is well researched and nicely written, therefore I believe that this paper does not need a further revision, therefore the manuscript meets the Journal’s high standards for publication.
I am always available for other reviews of such interesting and important articles.
Thank You for your work.